Human hair shaft proteomic profiling: individual differences, site specificity and cuticle analysis

Laatsch Chelsea N. 1
Durbin-Johnson Blythe P. 2
Rocke David M. 2
Mukwana Sophie 3
Newland Abby B. 4
Flagler Michael J. 4
Davis Michael G. 4
Eigenheer Richard A. 5
Phinney Brett S. 5
Rice Robert H. 1 rhrice@ucdavis.edu
1 Forensic Science Graduate Program and Department of Environmental Toxicology, University of California , Davis, CA , USA
2 Division of Biostatistics, Department of Public Health Sciences, Clinical and Translational Science Center Biostatistics Core, University of California , Davis, CA , USA
3 Biotech Forensics , Nairobi , Kenya
4 Procter & Gamble, Mason Business Center , Mason, OH , USA
5 Proteomics Core Facility, University of California , Davis, CA , USA
Uversky Vladimir
Electronic publication date: 2014 Aug 5
Publication date: 2014
Volume: 2
Electronic Location ID: e506
Received 2014 Jun 13; Accepted 2014 Jul 18
Copyright: © 2014 Laatsch et al.
Copyright year: 2014
Copyright holder: Laatsch et al.
License: This is an open access article distributed under the terms of the Creative Commons Attribution License, which permits unrestricted use, distribution, reproduction and adaptation in any medium and for any purpose provided that it is properly attributed. For attribution, the original author(s), title, publication source (PeerJ) and either DOI or URL of the article must be cited.
License URL: https://creativecommons.org/licenses/by/4.0/

Keywords: Ancestry, Body site, Corneocytes, Differential expression, Forensic evidence, Keratin associated proteins, Keratins, Proteomics, Transglutaminase

Funding: National Institutes of Justice 2011-DN-BX-K543 National Institute of Environmental Health Sciences 2 P42 ES04699 National Center for Advancing Translational Sciences (NIH) UL1 TR000002 This work was supported by National Institutes of Justice grant 2011-DN-BX-K543, National Institute of Environmental Health Sciences grant 2 P42 ES04699, the National Center for Advancing Translational Sciences (NIH) grant #UL1 TR000002 and a research contract from Procter & Gamble. The funders had no role in study design, data collection and analysis, decision to publish, or preparation of the manuscript.

==============================
Hair from different individuals can be distinguished by physical properties. Although some data exist on other species, examination of the individual molecular differences within the human hair shaft has not been thoroughly investigated. Shotgun proteomic analysis revealed considerable variation in profile among samples from Caucasian, African–American, Kenyan and Korean subjects. Within these ethnic groups, prominent keratin proteins served to distinguish individual profiles. Differences between ethnic groups, less marked, relied to a large extent on levels of keratin associated proteins. In samples from Caucasian subjects, hair shafts from axillary, beard, pubic and scalp regions exhibited distinguishable profiles, with the last being most different from the others. Finally, the profile of isolated hair cuticle cells was distinguished from that of total hair shaft by levels of more than 20 proteins, the majority of which were prominent keratins. The cuticle also exhibited relatively high levels of epidermal transglutaminase (TGM3), accounting for its observed low degree of protein extraction by denaturants. In addition to providing insight into hair structure, present findings may lead to improvements in differentiating hair from various ethnic origins and offer an approach to extending use of hair in crime scene evidence for distinguishing among individuals.

Introduction

Mass spectrometry coupled with database searching now permits identification of many proteins in complex structures such as hair shaft (Lee, Rice & Lee, 2006; Barthélemy et al., 2012). In addition to its high keratin content, hair is challenging to analyze in part due to the extensive transglutaminase-mediated isopeptide cross-linking that prevents solubilization of ≈15% of the constituent protein even by strong denaturants under reducing conditions. Since only ≈20% of the lysines participate in isopeptide bonding, generation of peptides by trypsin fragmentation permits efficient identification of the cross-linked as well as non-cross-linked protein and comparison of relative amounts in parallel samples. The cross-linked proteins, a sampling of those present in corneocytes, include those capable of being solubilized and are derived from the various cellular compartments.

The structure of the hair shaft is complex, with corneocytes of the cuticle, cortex and medulla exhibiting distinct features readily visible ultramicroscopically after extraction with strong protein denaturants such as sodium dodecyl sulfate (SDS) under reducing conditions (Rice, Wong & Pinkerton, 1994). For example, cuticle cell protein appears much less extractable than that of the cortex, attributed to a higher level of isopeptide bonding (Zahn et al., 1980). Subfractionation of the cuticle by physical and enzymatic methods demonstrated different amino acid compositions indicative of different protein compositions (Swift & Bews, 1974a). A comprehensive analysis of cuticle from wool fibers clarified that it contains abundant keratins and keratin-associated proteins (KAPs) as well as a variety of structural proteins and enzymes, similar to human hair (Koehn et al., 2010).

Previous analyses reported in the literature using hair clipped from inbred mouse strains suggest that humans might show differences among individuals or according to ancestry (Rice et al., 2009; Rice et al., 2012). This possibility is consistent with observations that the large majority of genes are differentially expressed among individual humans and that a smaller fraction varies among specific populations (Lappalainen & Dermitzakis, 2010). Present work explores whether a limited survey of hair samples from several ethnic groups is sufficient to reveal such variation, and whether hair shafts from the scalp are distinguishable from other anatomic locations. It also explores whether the cuticle has a distinctive protein profile compared to intact hair shaft.

Materials and Methods

Hair samples

Samples of scalp and body hair were collected from unrelated volunteer adult subjects (ages 20–65) with written consent approved by the University of California Davis Institutional Review Board (protocol 217868). Samples were collected from individuals whose hair was not chemically treated (dyed, bleached, straightened). Scalp hair from Caucasian (six male), African–American (five male), Korean (three male, two female) and Kenyan (four male, one female) individuals were analyzed. Three samples from each subject were analyzed; in most cases, scalp hair was collected at the same time and processed from three sites on the head (left, right, center). For analysis of cuticle, several grams of hair were collected from participating individuals (five Caucasian males, two Asian females) during a haircut for single analyses. The parents of Korean and Kenyan subjects were also Korean or Kenyan, respectively. Samples of axillary, facial (beard) and pubic hair were collected from male subjects.

Cuticle isolation

Hair samples were wetted with distilled water, rinsed with clarifying shampoo and then rinsed with water several times with gentle swirling. Initial samples (0.2 g) were transferred to glass scintillation vials, suspended in 2 ml of distilled water and shaken at 2,500 rpm in a DVX-2500 Multi-tube Vortexer (VWR, Radnor, PA). After various lengths of time, suspended cells were recovered from the water fraction by centrifugation. One aliquot was submitted for protein determination and another was examined by scanning electron microscopy after air drying and coating with Au/Pd in a Hitachi S-4700 instrument at 3 kV with built in PCI image software. Purity of the cuticle cells was estimated by visual inspection. Larger aliquots of hair were then processed together at the empirically determined optimal time period.

Sample processing

Samples (2–4 mg) were rinsed twice in 2% SDS–0.1 M sodium phosphate (pH 7.8) and incubated in 0.4 ml of this buffer containing 25 mM dithioerythritol for disulfide reduction and then alkylation with iodoacetamide. Proteins were recovered as a flocculent precipitate by centrifugation after addition of 1 ml of ethanol, rinsed twice with 67% ethanol, once with freshly prepared 0.1 M ammonium bicarbonate and digested for 3 days with daily additions of 40 µg of stabilized trypsin (Rice, Means & Brown, 1977). Clarified digests, containing 90% of the digested protein, were submitted for mass spectrometric analysis. Alkylation of the protein with 2-bromoethylamine instead of iodoacetamide provided marginally higher peptide yields for many proteins but largely suppressed identification of keratin-associated proteins as previously reported (Rice et al., 2012).

Mass spectrometry and protein identification

Samples adjusted to approximately equal peptide amounts by A280 and adjusted to 0.5% trifluoracetic acid were directly loaded onto an Agilent ZORBAX 300SB C18 reverse-phase trap cartridge which, after loading, was switched in-line with a Michrom Magic C18 AQ 200 µm × 150 mm nano-LC column connected to a Thermo-Finnigan LTQ iontrap mass spectrometer through a Michrom Advance Plug and Play nanospray source with CTC Pal autosampler. The nano-LC column was used with a binary solvent gradient; buffer A was composed of 0.1% formic acid and buffer B composed of 100% acetonitrile. The 120 min gradient consisted of the steps 2–35% buffer B for 85 min, 35–80% buffer B for 23 min, hold for 1 min, 80–2% buffer B for 1 min, then hold for 10 min, at a flow rate of 2 µl/min for maximal separation of tryptic peptides. An MS survey scan was obtained for the m/z range 375–1400, and MS/MS spectra were acquired from the 10 most intense ions in the MS scan by subjecting them to automated low energy CID. An isolation mass window of 2 Da was used for the precursor ion selection, and normalized collision energy of 35% was used for the fragmentation. A 2 min duration was used for the dynamic exclusion. Monitoring of column washes indicated negligible intersample contamination.

Tandem mass spectra were extracted with Xcalibur version 2.0.7. All MS/MS samples were analyzed using X!Tandem (The GPM, thegpm.org; version CYCLONE (2012.10.01.2)). X!Tandem was set up to search a 2012 Uniprot human database appended to a database of common non-human contaminants (cRAP, http://www.thegpm.org/crap/), both of which were appended to an identical but reversed database for calculating false discovery rates (136,252 proteins total), assuming the digestion enzyme was trypsin. X!Tandem was searched with a fragment ion mass tolerance of 0.40 Da and a parent ion tolerance of 1.8 Da. Iodoacetamide derivative of cysteine was specified in X!Tandem as a fixed modification. Deamidation of asparagine and glutamine, oxidation of methionine and tryptophan, sulfone of methionine and tryptophan oxidation to formylkynurenin of tryptophan were specified in X!Tandem as variable modifications. Scaffold version 4.2.1 (Proteome Software Inc., Portland, OR) was used to validate MS/MS based peptide and protein identifications. Peptide identifications were accepted if they could be established at greater than 90% probability as specified by the Peptide Prophet algorithm (peptide decoy false discovery rate 0.2%) (Keller et al., 2002). Protein identifications were accepted if they could be established at greater than 99% probability and contained at least two identified peptides (protein decoy false discovery rate 4.1%). Protein probabilities were assigned by the Protein Prophet algorithm (Nesvizhskii et al., 2003). Numbers of distributed spectral counts (called weighed spectral counts in Scaffold) were tabulated using experiment-wide grouping. The Scaffold file containing all the peptide data used in the analysis is now available in the public database on the MassIVE repository (ID: MSV000078650; http://massive.ucsd.edu).

Because certain keratins are well known to contain identical peptides, we used a distributed spectral count (called weighted spectral count in Scaffold) to model spectral counts more accurately across the proteins identified in this study. Scaffold’s weighed spectral counts appropriates a percentage of each count divided among the protein groups that share that peptide. The formulation for that percentage for peptide(i) assigned to protein(j) is PPS(j)/sum(PPS(1…n)), where PPS(j) is the sum of the peptides(1…m) for protein(j) and (1…n) is the set of proteins that contain peptide(i). The weighted spectrum count is the sum of those percentages for each protein group. This is similar to method 3-a in Zhang et al. (2010), but Scaffold uses the summed probabilities as normalizers instead of summed exclusive counts (B Searle, pers. comm., 2014). In addition, every protein identified was required to have at least one unique peptide that was not shared with any other protein.

Statistical analysis

Total hair and cuticle samples from different subjects were compared using mixed-effects overdispersed Poisson regression models, including a fixed effect for total hair vs. cuticle, a fixed effect for ethnicity (to adjust for imbalances between total hair and cuticle donors) and a random effect for subject. This analysis included scalp total hair samples from one Caucasian subject, five Kenyan subjects, five African American subjects, and five Korean subjects; cuticle samples were from five white subjects and two Asian subjects.

Hair from different sites (scalp, axillary, facial, and pubic) were likewise compared within subjects using mixed-effects overdispersed Poisson regression models, including a fixed effect for site and a random effect for subject. Pairwise comparisons between sites were conducted using the Tukey HSD method. This analysis included scalp, axillary, facial, and pubic hair samples from four Caucasian subjects.

Scalp samples were compared among subjects of the same ethnicity using an overdispersed Poisson model, treating subject as a fixed effect.

To include proteins expressed at very low levels, a different analysis approach was taken for the comparison of hair and cuticle samples within subjects. Normalization factors were calculated using the Trimmed Mean of M-Values (TMM) method developed for RNA-Seq data (Robinson & Oshlack, 2010). Counts were then divided by these normalization factors. These preprocessing steps were conducted once for the entire data set and not repeated for individual analyses.

Total hair and cuticle samples were then compared within subjects using mixed-effects lognormal-Poisson regression models (similar to negative binomial models), including a fixed effect for total hair vs. cuticle and a random effect for subject. This analysis included scalp total hair and cuticle samples from five Caucasian subjects. Mixed effects lognormal-Poisson regression modeling was conducted using the lme4 package, version 0.999999-2 (Bates, Maechler & Bolker, 2013) in the statistical software environment R, version 3.0.1 (R Core Team, 2013).

Error bars in figures represent 95% confidence intervals for parameter estimates from the models described above. Hierarchical clustering was performed with the hclust function in the R statistical software environment using the complete linkage method as described in http://nlp.stanford.edu/IR-book/html/htmledition/single-link-and-complete-link-clustering-1.html (Manning, Raghavan & Schütze, 2008).

Results

Distinguishing among hair shaft samples within ethnic groups

Scalp hair samples from Caucasian, African–American, Korean and Kenyan individuals were analyzed. Analysis of triplicate samples from each subject (one each from different regions of the scalp) provided assurance that the differences among samples reflected individual subject differences. Considerable variation was evident in the profiles of prevalent proteins within each ethnic group. Using pairwise comparisons of 76 proteins present in sufficient amounts, each group was analyzed separately (Table S2–Table S5). As shown in Table 1, samples from individuals were distinguishable in each group, although the number of significant protein differences varied greatly. Fig. 1 displays for the Caucasian group nine proteins that were present at significantly different levels in comparisons among at least some of the 6 individuals analyzed. In such analyses, K32, K33B, K34 and K83 exhibited significant differences in at least 40% of the pairwise comparisons within all four ethnic groups, while K31, K35, K39 and K86 exhibited similarly high inter-subject variability in at least three of the ethnic groups.

Figure 1 Weighted spectral counts for nine proteins differentially expressed in hair shafts from six Caucasian subjects.

Samples were analyzed separately from three locations on the scalp.

Table 1 Pairwise comparisons of 76 proteins in profiles from samples in each of four ethnic groups.

Values are number of proteins that were significantly different in weighted spectral counts. Subjects in each group—six Caucasian and five each African–American (African–Amer), Kenyan and Korean—are numbered and labeled CA, AA, KE and KO, respectively.

Caucasian	CA2	CA3	CA4	CA5	CA6	
CA1	7	5	11	15	26	
CA2		4	16	13	27	
CA3			14	23	23	
CA4				23	34	
CA5					30	
African–Amer	AA2	AA3	AA4	AA5	
AA1	26	22	24	18	
AA2		18	20	20	
AA3			5	6	
AA4				9	
Kenyan	KE2	KE3	KE4	KE5	
KE1	6	14	13	16	
KE2		10	5	21	
KE3			22	22	
KE4				18	
Korean	KO2	KO3	KO4	KO5	
KO1	2	3	37	39	
KO2		4	40	40	
KO3			28	32	
KO4				21	

Distinguishing samples from different ethnic groups

In contrast to the high variation in protein expression within ethnic groups, consistent differences among ethnic groups were less marked. As shown in Table 2, the groups were distinguishable (Table S6), with African–American samples being the most distinctive. Most useful for pairwise comparisons were KAPs, accounting for 21 of the 32 significant differences (66%) observed overall. Illustrated in Fig. 2, KAPs 2-4, 4-3, 13-1 and 13-2 accounted for 16 (50%), whereas K40, selenium binding protein-1 (SBP1) and epidermal transglutaminase (TGM3) accounted for seven (22%) of the total significant differences.

Figure 2 Weighted spectral counts for six proteins differentially expressed in hair from subjects of different ethnic origin.

Each bar represents the aggregate for Caucasian (CA), African–American (AA), Kenyan (KE) and Korean (KO) samples.

Table 2 Pairwise comparisons of proteins in hair from subjects from different ethnic groups.

Values are significant differences in weighted spectral counts from hair from African–American (AA), Caucasian (CA), Kenyan (KE) and Korean (KO) subjects.

	CA	KE	KO	
AA	8	6	7	
CA		4	3	
KE			4	

Distinguishing hair shafts from different body sites

Hair samples from three body sites (axillary, facial and public regions) were analyzed and compared to those from scalp. As seen in Fig. 3, pairwise comparisons using 92 proteins permitted distinguishing among them (Table S7). Scalp hair displayed the most differences from the others, while axillary and pubic hair displayed the fewest differences from each other. In these comparisons, KRTs and KAPs together and in equal amounts accounted for only one-third of the significant differences. A variety of enzymes and structural proteins contributed to the observed differences as illustrated in Fig. 4.

Figure 3 Pairwise comparisons of proteins (92) in hair obtained from different body sites of the same four individuals.

The values shown for significantly different weighted spectral counts were used for hierarchical clustering by relatedness.

Figure 4 Weighted spectral counts for 10 proteins differentially expressed in hair from different body sites of Caucasian subjects.

Illustrated are composite values for single samples of Ax (axillary), Fa (facial), and Pu (pubic) hair from each of four Caucasian subjects and for triplicate values of Sc (scalp) hair from the same four subjects.

Cuticle analysis

The yield of cuticle cells from hair fibers increased with time of agitation in water as previously described (Swift & Bews, 1974b). After several hours, however, the purity of the released cuticle material was observed to decline as fragments from other parts of the hair shaft increased in amount. Since the optimal time varied among subjects, treatment of each sample was monitored by scanning electron microscopy to ensure its quality. The yield of purified cuticle cells was estimated as 0.1% of the total hair mass. Samples submitted for proteomic analysis were estimated to be >95% cuticle cells, with only rare non-cuticle cells being distinguishable, as illustrated in Fig. 5.

Figure 5 Example of cuticle fractions isolated for proteomic analysis.

(A) shows a sample of purity >95% cuticle cells used in the analysis, while (B, not used) had an estimated purity of 70%. The vertical white hash marks at the bottom of each panel mark 10 µm intervals.

The protein profiles of cuticle samples were distinctly different from those of the total hair. When samples of both were compared from five Caucasian subjects, 34 proteins were seen to be differentially expressed (Table S8A). An additional comparison was made between the cuticle samples from five Caucasian and two Asian subjects with total hair from Caucasian, Korean, Kenyan and African–American subjects. In that case, 30 proteins were seen to be differentially expressed (Table S8B). Of these, 24 overlapped in the two comparisons and were taken to be the most reliable indicators of differential expression. Of these, more than half were KRTs, with six higher (K1, K2, K10, K32, K40, K82) and eight lower (K31, K33A, K33B, K38, K39, K83, K85, K86) in the cuticle. Cuticle also displayed an elevated level of TGM3, consistent with the relatively low degree of protein extraction by detergent compared to the cortex (Rice, Wong & Pinkerton, 1994). Other proteins enriched in cuticle were AIM1, TUBB2A and VSIG8, while DSG4, DSP, KAP3-1, HIST1H2AG, HIST1H2BK and SFN were higher in total hair shaft. Fig. 6 illustrates 16 of the most distinctive proteins.

Figure 6 Differentially expressed proteins.

Illustrated are eight proteins whose weighted spectral counts are significantly higher in cuticle (Cut), shown in (A), and eight with higher levels in the total hair shaft (Tot) shown in (B). These compare levels in the same five subjects.

Discussion

Human corneocyte protein profiling, as conducted in this work, has provided insight into phenotyping the skin disease ichthyosis (Rice et al., 2013) and understanding the toxic response of cultured keratinocytes to the environmental toxicant 2,3,7,8-tetrachlorodibenzo-p-dioxin (Hu et al., 2013). Extending such efforts to compare hair shaft protein profiles among individuals has several possible applications. As previously demonstrated, the profiles can reflect structural features that may be pertinent to hair phenotype (Rice et al., 2009; Rice et al., 2012). This approach could now be extended specifically to the cuticle in cases of anomalous structure of that layer (Rice et al., 1999a). Monitoring of hair protein profiles could be informative with respect to keratin-related disease (Schweizer et al., 2007; Moll, Divo & Langbein, 2008) and, since the hair shaft exhibits a number of proteins with important functions elsewhere in the body, it could have more general diagnostic utility. In such work, knowledge of the range of individual variability will be valuable in analyzing profiling datasets. The relative paucity of proteins distinguishing hair from different ethnic origins suggests that some visible differences (e.g., curliness) are not bestowed by major structural proteins, and the importance of lipid processing for features such as combability is now appreciated (Shimomura et al., 2009).

An ability to distinguish individuals by means of hair could be applicable to crime scenes where this is a commonly recovered type of evidence. Microscopic examination of hair evidence can take into account features such as pigmentation (granule distribution and density, spectral analysis), cosmetic treatment (dye, bleach), diameter, appearance in cross-section and structural abnormalities of the shaft. Using all these features permits useful discrimination whether a given hair originates from a specific individual (Gaudette, 1999), leading to the recommendation that hair analysis should begin with a thorough microscopic examination (Lanning et al., 2009). Nevertheless, the search for more objective criteria by which to judge hair matches continues (Taupin, 2004). Hair evidence is ordinarily supplemented by DNA evidence whenever possible (Rowe, 2001). When follicle tissue is present, nuclear DNA extracted from a hair sample may identify the donor. In the great majority of cases, only mitochondrial DNA (mtDNA) from the shaft is available. mtDNA can provide valuable exclusionary evidence, but is not sufficient alone for individual identification. The information from proteomic characterization is complementary to that from microscopic examination and DNA analysis and may assist in discerning its body site of origin, thereby augmenting its evidentiary value. In present work, samples were subjected to pairwise comparison using expression levels of specific protein components. Further effort may permit development of a statistical classification scheme important in establishing a searchable database.

The substantial variation in inter-individual hair protein expression levels hypothesized was clearly manifest in present results. The major hair shaft components, KRTs and KAPs, were both useful in discriminating among individuals, but the latter appeared to be more useful among ethnic groups. A previous study also pointed to differences in KAPs among subjects of African, Jamaican and African–American origin (Porter et al., 2009). These findings, and the frequent length polymorphisms they display (Fujikawa et al., 2012), often reflecting amino acid repeats in the coding region, may impel further focus on KAPs for this purpose.

A strong genetic component appears to be involved in expression differences observed in other species. This likely reflects variation in transcription factor binding affinity for chromatin that can even have epigenetic consequences (Kasowski et al., 2013; Kilpinen et al., 2013; McVicker et al., 2013). However, indications that the profile in humans could depend on age (Giesen et al., 2011) and the possible influence of other factors including environment and diet (Almeida et al., 2014) merit further study. In addition, how well cuticle cells remain attached to the shaft could influence the profile. The latter could depend on hair length, weathering, and cuticle cell stability, variables among individuals (Rice et al., 1999a). These factors would need clarification in determining the value that a database of individual hair profiles would have for forensic science. Nevertheless, results so far provide a basis for further investigation.

TGM3 is known to be expressed in the cuticle and cortex of human hair shaft (Thibaut et al., 2005). Emphasizing that the cuticle exhibits intriguing differences in protein profile from cortex (Koehn et al., 2010), present results provide a rationale for its proposed high content of isopeptide bonding in its relatively high expression level of TGM3. This could contribute to the persistence of clearly demarcated endocuticle, exocuticle and marginal band (A) layers visible even after detergent extraction (Rice, Wong & Pinkerton, 1994), and is consistent with the severe structural defects (e.g., easier cuticle loss) due to TGM3 ablation (John et al., 2012). Not detected at such high levels in present work, TGM1 is known to be important for stabilization of human cuticle cells, since detergent extraction greatly perturbs their structure in individuals lacking this activity (Rice et al., 1999b).

Present results are complementary to exquisite immunochemical and in situ hybridization studies of keratin expression in various layers of the hair follicle (Moll, Divo & Langbein, 2008; Langbein et al., 2010). They permit identification of a variety of such proteins and others that are not KRTs and KAPs, without the need for isoform-specific antibodies, and are applicable to the mature hair shaft instead of the living cells. Similar to results with wool cuticle (Koehn et al., 2010), the analysis revealed several epidermal keratins including K1 and K10 previously identified in human hair cuticle (Stark et al., 1990). This unexpected finding appears to be in contrast to the above reviews concerning hair follicle, but is difficult to attribute to contamination of the purified preparations or to mis-identification due to peptides shared with other keratins. Further investigation may help resolve this dichotomy.

Supplemental Information

Table S1 Profiles of protein constituents in samples of hair and isolated cuticle

Click here for additional data file.

Table S2 Pairwise comparisons among Caucasian samples

Click here for additional data file.

Table S3 Pairwise comparisons of African–American samples

Click here for additional data file.

Table S4 Pairwise comparisons of Kenyan samples

Click here for additional data file.

Table S5 Pairwise comparisons of Korean samples

Click here for additional data file.

Table S6 Pairwise comparisons of samples of different ethnic origin

Click here for additional data file.

Table S7 Pairwise comparisons of samples from different body sites among Caucasian subjects

Click here for additional data file.

Table S8 Statistical testing of cuticle versus total hair shaft. Cuticle from 5 Caucasian subjects compared to scalp hair from the same subjects

Click here for additional data file.

We thank Dr. Ying L. Boissy for performing scanning electron microscopy.

Additional Information and Declarations

Competing Interests

Author Contributions

Human Ethics

Data Deposition

Sophie Mukwana is an employee of Biotech Forensics. Abby B. Newland, Michael J. Flagler and Michael G. Davis are employees of Procter & Gamble. Brett S. Phinney is an Academic Editor for PeerJ. The authors declare there are no competing interests.

Chelsea N. Laatsch and Robert H. Rice conceived and designed the experiments, performed the experiments, analyzed the data, wrote the paper, prepared figures and/or tables, reviewed drafts of the paper.

Blythe P. Durbin-Johnson analyzed the data, contributed reagents/materials/analysis tools, wrote the paper, prepared figures and/or tables, reviewed drafts of the paper.

David M. Rocke analyzed the data, contributed reagents/materials/analysis tools, reviewed drafts of the paper.

Sophie Mukwana contributed reagents/materials/analysis tools, reviewed drafts of the paper.

Abby B. Newland conceived and designed the experiments, performed the experiments, contributed reagents/materials/analysis tools, prepared figures and/or tables, reviewed drafts of the paper.

Michael J. Flagler conceived and designed the experiments, performed the experiments, contributed reagents/materials/analysis tools, reviewed drafts of the paper.

Michael G. Davis conceived and designed the experiments, contributed reagents/materials/analysis tools, reviewed drafts of the paper.

Richard A. Eigenheer performed the experiments, contributed reagents/materials/analysis tools, reviewed drafts of the paper.

Brett S. Phinney performed the experiments, analyzed the data, contributed reagents/materials/analysis tools, wrote the paper, reviewed drafts of the paper.

The following information was supplied relating to ethical approvals (i.e., approving body and any reference numbers):

University of California Davis Institutional Review Board (protocol 217868).

The following information was supplied regarding the deposition of related data:

MassIVE repository (ID: MSV000078650; http://massive.ucsd.edu).

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
