# Peer review of "Human hair shaft proteomic profiling: individual differences, site specificity and cuticle analysis"

_PeerJ, doi:10.7717/peerj.506_

## Round 0.1 · original submission · Minor Revisions

Please address all the points raised by both reviewers.

Reviewer 1 ·

Basic reporting

No Comments

Experimental design

1) The age of volunteers is not specified in materials and methods (hair samples), it is unclear whether there were there individuals with stained hair or any disturbances in hair shaft structure among them
2) In «cuticle isolation» section, these is a mistake: authors possibly meant «air drying» instead of «air dying»
3) In my opinion, the work is not enough discussion of the differences in the structure of the hair shaft.
For example the intact hair pictures, made with in particular an scanning electron microscope (authors own samples and method), would attract a wider range of researchers. I wonder whether there are differences in the pattern of the cuticle (or other characteristics of the hair shaft, at the discretion of the authors) in hair from different body sites and scalp hair of different ethnic groups (Caucasian, African-American, Kenyan and Korean subjects). If there are clear differences persist in samples from each individual of the ethnic group, it complements and enhances the definition of hairs of different ethnic groups. Especially, the authors showed that «differences between ethnic groups less marked».
Description of the differences in the structure of the hair shaft (physical properties, morphology, cuticle pattern) between the hair from different body sites (axillary, facial pubic) could serve as an additional rapid method for forensic evidence.

Validity of the findings

Overall, the present study can be interesting and demanded both in proteomics, and for studies of skin diseases.

Reviewer 2 ·

Basic reporting

The English is ok, but in some places is a little ornate, rather than straightforward. The authors should review this before publication. For example in the abstract- levels of a score of proteins- do they mean 20!

Line 247 frequent length polymorphism could be explained a bit better.

Experimental design

Looks good

Validity of the findings

A reasonable number of samples have been examined. In may have been that if a larger number of samples had been examined more differences would be seen between ethic groups for example.

I did not think Figure 3 was all that novel. The clustering is as one might expect.

Additional comments

None

---

## Round 0.2 · accepted · Accept

I appreciate your careful consideration of the reviewers' comments and comment on the great job you did revising your manuscript.